# Circulating Tumor DNA in Precision Oncology and Its Applications in Colorectal Cancer

**DOI:** 10.3390/ijms23084441

**Published:** 2022-04-18

**Authors:** Maria F. Arisi, Efrat Dotan, Sandra V. Fernandez

**Affiliations:** 1Sidney Kimmel Medical School, Thomas Jefferson University, Philadelphia, PA 19107, USA; mfa007@jefferson.edu; 2Department of Medical Oncology, Fox Chase Cancer Center, Philadelphia, PA 19111, USA; efrat.dotan@fccc.edu; 3Department of Pathology, Fox Chase Cancer Center, Philadelphia, PA 19111, USA

**Keywords:** circulating tumor DNA, cell-free DNA, biomarkers, minimal residual disease, MRD, clonal hematopoiesis, CHIP, colorectal cancer, CRC

## Abstract

Circulating tumor DNA (ctDNA) is a component of cell-free DNA (cfDNA) that is shed by malignant tumors into the bloodstream and other bodily fluids. ctDNA can comprise up to 10% of a patient’s cfDNA depending on their tumor type and burden. The short half-life of ctDNA ensures that its detection captures tumor burden in real-time and offers a non-invasive method of repeatedly evaluating the genomic profile of a patient’s tumor. A challenge in ctDNA detection includes clonal hematopoiesis of indeterminate potential (CHIP), which can be distinguished from tumor variants using a paired whole-blood control. Most assays for ctDNA quantification rely on measurements of somatic variant allele frequency (VAF), which is a mutation-dependent method. Patients with certain types of solid tumors, including colorectal cancer (CRC), can have levels of cfDNA 50 times higher than healthy patients. ctDNA undergoes a precipitous drop shortly after tumor resection and therapy, and rising levels can foreshadow radiologic recurrence on the order of months. The amount of tumor bulk required for ctDNA detection is lower than that for computed tomography (CT) scan detection, with ctDNA detection preceding radiologic recurrence in many cases. cfDNA/ctDNA can be used for tumor molecular profiling to identify resistance mutations when tumor biopsy is not available, to detect minimal residual disease (MRD), to monitor therapy response, and for the detection of tumor relapse. Although ctDNA is not yet implemented in clinical practice, studies are ongoing to define the appropriate way to use it as a tool in the clinic. In this review article, we examine the general aspects of ctDNA, its status as a biomarker, and its role in the management of early (II–III) and late (IV; mCRC) stage colorectal cancer (CRC).

## 1. Cell-Free DNA and Circulating Tumor DNA

Cell-free DNA (cfDNA) is comprised of double-stranded DNA ranging between 150 and 200 base pairs in length and circulates mostly in blood [1]. Passive release via apoptosis, necrosis, and phagocytosis account for the primary mechanisms of cfDNA release. In healthy individuals, most of the cfDNA originates from hemopoietic cells such as erythrocytes, leukocytes, and endothelial cells [2]. Normal tissue that undergoes damage by ischemia, trauma, infection, or inflammation can also release cfDNA [3]. Active secretion via exosomes or protein complexes also contributes to cfDNA [4]. cfDNA has a short half-life ranging between 16 min and 2.5 h, although it may be longer when it is bound to protein complexes or inside membrane vesicles [5,6], and it is cleared from circulation via nuclease action [7] and renal excretion into the urine [8].

Circulating tumor DNA (ctDNA) is a component of cfDNA that is shed by malignant tumors into the bloodstream and other bodily fluids, including blood, urine, pleural fluid, ascites, and saliva (Figure 1). In the case of brain tumors, including brain metastatic lesions, ctDNA can also be found in the cerebrospinal fluid [9,10]. ctDNA typically constitutes a small proportion of an individual’s total cfDNA [11]. Plasma samples are preferable to serum for ctDNA analysis as the latter contain larger quantities of DNA from leukocytes lysed during the clotting process, thereby increasing the background to signal ratio and interfering with ctDNA detection [12]. Compared to cfDNA derived from non-cancer cells, ctDNA is shorter [13,14,15]. cfDNA from normal cells has a fragment size of ~166 bp, and ctDNA has a fragment size of ~146 bp because of the loss of the H1 linker [16]. Plasma ctDNA is also generally more fragmented than non-mutant cfDNA, with a maximum enrichment between 90 and 150 bp, as well as enrichment in the size range 250–320 bp [16].

The amount of cfDNA in cancer patients varies widely, and is usually higher than in the blood of healthy controls [17]. In healthy donors, the mean value of plasma cfDNA is ~10.3 ng/mL plasma (range: 5–50 ng/mL plasma), with a median value of 5 ng/mL plasma [18,19]. cfDNA in cancer patients is usually below 100 ng/mL plasma and approximately below 17,000 genome equivalent (GE) per ml of plasma (assuming 6 pg of DNA per diploid human genome) [17]. The concentration of ctDNA can exceed 10% of the total cfDNA in patients with advanced-stage cancers [20] but is much lower in patients with low tumor burden, such as patients with localized disease [20,21,22]. In colon cancer, the concentration of ctDNA in pretreatment plasma is significantly lower in stage I patients compared with stage II–III patients [23] and the ctDNA fraction of the total cfDNA is often less than 0.1% in patients with early stage colon cancer following curative surgery [21]. Varying ctDNA levels are associated with clinical and pathological features of cancer, including stage, tumor burden, localization, vascularization, and response to therapy [11,20,24,25,26]. Furthermore, ctDNA levels vary due to differences in tumor grade (e.g., indolent vs. fast progressing), shedding rates, and other biological factors [27]. Bettegowda et al. reported how ctDNA detectability varies among different tumor types [20]. Most patients with stage III ovarian and liver cancers and metastatic cancers of the pancreas, bladder, colon, stomach, breast, esophagus, and head and neck, as well as patients with neuroblastoma and melanoma, harbored detectable levels of ctDNA [20]. In contrast, less than 50% of patients with medulloblastomas or metastatic cancers of the kidney, prostate, or thyroid, and less than 10% of patients with gliomas, harbored detectable ctDNA [20].

In one study of colorectal cancer (CRC) patients, the mean value of cfDNA was 495.7 ng/mL of plasma (range 100–1750 ng/mL), with a median value of 450 ng/mL [18]. Four months after primary resection, CRC patients had a significant decrease in plasma cfDNA levels to a mean of ~170 ng/mL (range 15–500 ng/mL), with a median value of 110 ng/mL plasma [18]. In CRC patients with tumor recurrence, a dramatic increase in plasma cfDNA was observed, while in disease-free patients, a continuous decrease was seen [18]. A significant association between estimated disease volume based on computerized tomography (CT) and the ctDNA fraction was found in metastatic breast cancer (mBC) and non-small cell lung cancer (NSCLC) patients [28]. Similarly, in castration-resistant prostate cancer (CRPC) patients who had dominant bone metastases, there was an association between the automated bone scan index and the ctDNA fraction [28]. ctDNA levels and disease volume assessed by imaging were also significantly correlated in patients with relapsed high-grade serous ovarian cancer [29]. ctDNA provides a more sensitive method of detecting malignancies than imaging or other conventional approaches (Figure 1). ctDNA showed incredible potential as a highly sensitive and specific cancer biomarker [25,26]. Various studies in breast, lung, and colorectal cancers demonstrated the potential clinical applications of ctDNA analysis at each stage of cancer management: molecular profiling [30,31,32,33,34], prognosis and staging [20,35,36], the detection of minimal residual disease (MRD) [37,38], monitoring response to therapy (post-treatment tumor surveillance), and clonal evolution [39,40,41,42,43].

Two types of information can be obtained through ctDNA analysis: genomic sequences in tumors and the quantification of tumor burden. Quantifying ctDNA at a single time point may allow disease staging and prognosis, and sequence analysis might allow the selection of targeted therapies. ctDNA carries genetic information from the primary and metastatic lesions and can therefore provide insights into clonal heterogeneity and the evolution of the tumor [30]. In contrast to the analysis of tumor biopsies, which are invasive to obtain and often do not fully capture tumor heterogeneity and evolution, the analysis of ctDNA offers a non-invasive method of repeatedly evaluating the genomic profile of a patient’s tumor [44]. For ctDNA quantification, the optimal methodology and modality for quantification has not been determined; most assays for ctDNA quantification rely on measurements of somatic variant allele frequency (VAF), which is a mutation dependent method. ctDNA assays must detect mutations in plasma at allele fractions of <0.1% to achieve that goal [45]. However, a proportion of patients might not have detectable somatic variants in ctDNA due to a low tumor burden, limited assay sensitivity, or the true absence of detectable somatic alterations. Current strategies to improve ctDNA detection rely on increasing the depth of sequencing coupled with various error-correction methods [25,46].

## 2. Use of ctDNA in Early Stage Disease

Most of the early data about ctDNA applications in oncology were in the metastatic solid tumor setting for molecular profiling at diagnosis, targeted therapy selection, treatment response monitoring, post-treatment tumor surveillance to detect recurrence, and monitoring treatment resistance [47] (Figure 2). However, there are currently several applications at early disease stages; ctDNA can potentially be used in early stage solid tumor patients at diagnosis for molecular profiling when tumor tissue is not available, as a biomarker for response to neoadjuvant and adjuvant therapies, to detect MRD postoperatively, and to guide treatment selection (Figure 2).

In the neoadjuvant and adjuvant settings, measuring serial ctDNA prior to and throughout treatment may be useful to monitor response to treatment as an early endpoint to potentially predict long-term outcomes [48]. In early stage breast cancer patients’, the clearance of ctDNA was a predictor of a pathological complete response to neoadjuvant treatment and was associated with a lower risk of recurrence [49]. During adjuvant chemotherapy, a serial ctDNA collection in patients with resected colon cancer found that increases in ctDNA levels during treatment were an early indicator of radiological recurrence and could be an early predictor of relapse [21,23]. Evidence is emerging on the potential to detect MRD by ctDNA assessment post-surgery to guide decisions on adjuvant therapy. A study of patients with operable urothelial cancer found that the presence of ctDNA after surgery was significantly associated with poor prognosis, and those with detectable ctDNA appeared to derive the most relative benefit from adjuvant immunotherapy [50].

Data demonstrated that the amount of tumor bulk required for ctDNA detection is lower than for CT scan detection, with ctDNA detection preceding radiologic recurrence in many cases [21]. This sensitivity can be exploited in several ways. One potential use is the early diagnosis of recurrent cancer prior to the emergence of clinical or radiological manifestations and in the detection of MRD, defined as the detection of ctDNA with no other clinical evidence of disease recurrence in patients who have completed all potentially curative therapies [51]. In patients with radiographically evident disease, ctDNA also seems to be more sensitive to changes in tumor burden and might assist in tailoring the intensity of therapy in the neoadjuvant setting and in monitoring for tumor response in patients requiring palliative treatment [51].

In the early stage disease setting, adjuvant trial patient populations are heterogeneous, and several clinical trials are evaluating the potential value of ctDNA to select patients that could benefit from further treatments. The MERMAID-1 (NCT04385368) and MERMAID-2 (NCT04642469) clinical trials are assessing adjuvant treatment with a checkpoint inhibitor in patients with resected stage II and III NSCLC with MRD positive by ctDNA measurements. IMvigor 011 (NCT04660344) for patients with muscle-invasive bladder cancer who are ctDNA positive after cystectomy is exploring the use of ctDNA as a biomarker for patient selection for adjuvant treatment with anti-PDL1 antibodies. These studies are evaluating the use of ctDNA to identify patients at the highest risk of recurrence for enrollment in clinical trials, reducing patient numbers as well as the time and cost of studies.

## 3. Use of ctDNA to Detect Minimal Residual Disease

The detection of ctDNA following surgery or treatment with curative intent may signal the presence of MRD, which could identify patients who may be at a higher risk of relapse (Figure 2). Analysis of ctDNA showed a promising clinical potential to detect MRD for solid tumors following treatment and in advance of radiological disease relapse [22]. The presence of ctDNA containing somatic mutations found in an individual’s tumor is a direct indication that occult tumor cells remain after surgery. It is only when the primary tumor is removed that the presence of ctDNA indicates residual micrometastatic disease, which is associated with the risk of recurrence. MRD detection by ctDNA profiling correlated with worse prognosis in patients with different types of solid tumors [21,22,38,52,53,54]. Due to the low ctDNA concentration associated with MRD, detection techniques must be sensitive enough to detect mutations in plasma at a VAF of <0.1% [54].

Chaudhuri et al. [22] showed that in patients with localized lung cancer, ctDNA detected within 4 months of treatment completion identifies patients who eventually relapsed by the detection of multiple somatic mutations, both driver and passengers. In this study, ctDNA levels at MRD were detected to VAF as low as 0.003% [22], and ctDNA-positive patients had worse overall survival (OS) compared with ctDNA-negative patients (*p* < 0.001, hazard ratio [HR] 14.3, 95% confidence interval [CI] 3.2–64.1) [22]. Furthermore, post-treatment ctDNA detection preceded radiographic progression by a median of 5.2 months in 72% of patients who developed recurrence [22]. In pancreatic cancer, liquid biopsy analyses demonstrate that 43% of patients with localized disease have detectable ctDNA at diagnosis, and detection of ctDNA after resection predicts clinical relapse and poor outcome, with recurrence by ctDNA, detected 6.5 months earlier than with CT imaging [53].

In addition, ctDNA analysis could guide patient-directed therapy, particularly in those patients in whom it is difficult to obtain a tumor biopsy. ctDNA sequencing enables the tumor-specific molecular profile of the patients to guide targeted therapy for precision medicine. Some patients in the study by Chaudhuri et al. had activating EGFR mutations detectable in ctDNA that might have enabled earlier targeted intervention with first-generation EGFR tyrosine kinase inhibitors, such as erlotinib [22].

## 4. Platforms for ctDNA Detection

Droplet digital PCR (ddPCR) and BEAMing (beads, emulsion, amplification, and magnetics) are two of the most prevalent digital PCR-based methods for the assessment of ctDNA and the most sensitive techniques currently available for detecting known mutations present in plasma ctDNA [55,56]. Both BEAMing and ddPCR have high sensitivity with a VAF for detection ≤0.01% [55,57,58,59]. However, both platforms have a limit on detecting a small number of mutations and are unable to detect mutations not known a priori, thus limiting the ability to address the issues related to intratumor heterogeneity and emergent mutations [1].

ddPCR uses water-in-oil droplets taking advantage of simple microfluidic circuits and surfactant chemistries to divide a 20 µL mixture of sample and reagents into ~20,000 monodisperse droplets (i.e., partitions) [60]. These droplets support PCR amplification of single template molecules using homogeneous assay chemistries and workflows similar to those widely used for real-time PCR applications (i.e., TaqMan). An automated droplet flow-cytometry with two-color detection reads each set of droplets after PCR, and droplets are assigned as positive or negative based on their fluorescence amplitude [60]. The number of positive and negative droplets in each channel is used to calculate the concentration of the target and reference DNA sequences. As the droplet volume is known, the fraction of positive droplets is then used to calculate the absolute concentration of the target sequence [60].

BEAMing combines emulsion PCR with flow cytometry, which makes it a powerful tool. BEAMing is a process built on emulsion PCR that includes beads within the compartments and ensures that one strand of the PCR product is bound to the beads [61]. After amplification, each compartment contains a bead that is coated with thousands of copies of the single DNA molecule originally present. These beads can be recovered with a magnet or by centrifugation and can be analyzed using flow cytometry or optical scanning instruments [62]. BEAMing involves a preamplification step of starting the DNA template with specific primers targeting the genomic loci of interest. The products are then put into a limiting dilution with primer-coated beads and undergo further PCR reaction before the beads are purified and attached to allele-specific fluorophore probes to delineate mutant from wild type. In contrast, in ddPCR, the initial template is subjected to a limiting dilution before any PCR [56]. Thus, for BEAMing, the entire starting template for all tested mutations is present in the initial reaction, although a threshold must be applied to exclude PCR error [56]. With ddPCR, the total DNA template is split for the different multiplexes.

O’Leary et al. [56] compared BEAMing and ddPCR for detecting mutations in ctDNA. Baseline plasma samples from patients with advanced breast cancer enrolled in the phase 3 PALOMA-3 trial were assessed for *ESR1* and *PIK3CA* mutations in ctDNA using both techniques. This large, clinically relevant comparison (*n* = 363 patients) showed good agreement between BEAMing and ddPCR [56]. ddPCR and BEAMing assays tailored to detect a tumor’s somatic mutations (known from prior next-generation sequencing (NGS) may identify MRD in patients at higher risk of relapse with great accuracy [21,22,38]. Conventional quantitative PCR (qPCR or real-time PCR) was used to study ctDNA in several cancer types, including CRC [63], although qPCR has lower sensitivity compared to digital PCR-based methods. The qPCR sensitivity is 0.1–1% [64,65]. It is well established that digital PCR technologies inherently provide greater sensitivity (~0.02–0.01%) than qPCR techniques for somatic mutation detection [64,66]. Mass-spectrometry based method is also used for ctDNA detection and is an adaptation of the conventional PCR method with multiplex detection. This method, such as UltraSEEK (Agena Bioscience, San Diego, CA, USA), first applies multiplex PCR to amplify and then mutations are captured with the labeled chain terminators for sing-base extension and identified using matrix-assisted laser desorption/ionization time-of-flight mass spectrometry [59,67]. Although some PCR-based methods are very sensitive and less expensive, they can only screen for known variants, and the input and speed are limited.

NGS-based assays are also used to study ctDNA and can assess mutations across broad areas of the genome not limited to known mutations [25]. Unique molecular identifiers or unique barcodes can help to increase sensitivity and specificity [45,68]. NGS can be applied using targeted panels for the specific detection of certain mutations. Among the NGS-based assays, targeted sequencing platforms, including Tagged-Amplicon deep sequencing (TAm-Seq) [69], Safe-Sequencing System (Safe-SeqS) [70], and Cancer Personalized Profiling by deep sequencing (CAPP-Seq) [46,71] have detection limits as low as 0.01% but are more expensive and time-consuming than PCR based methods. The optimal panel of genetic mutations for NGS-based assays would depend on the objective of the study; the detection of resistance mutations would require high sensitivity and high coverage, while monitoring will focus more on the specificity of given mutations [72].

One major challenge in detecting MRD from peripheral blood is that a typical blood draw sample only contains a few thousand copies of each gene. This means that when tracking one or few mutations, one will be unable to detect MRD when the fraction of cancerous cfDNA in the blood is lower than twice the inverse of the number of copies of each gene in a given sample (called the genomic equivalent limit (GEL) [73]. While collecting more blood may not always be feasible, tracking many mutations per patient increases the statistical likelihood that cfDNA fragments containing the desired targeted mutations are captured when tumor fraction in the blood is lower than GEL [46]. It was shown that tracking large numbers of individualized tumor mutations in cfDNA increases the detection rate of MRD [73]. One approach used to detect MRD is to apply targeted NGS using a 200–500 genes panel (or whole-exome sequencing; WES) to define up to several hundred mutations from each patient’s tumor and then develop an assay to track these mutations in blood cfDNA (Figure 3A) [73]. Parsons et al. found that tracking a customized fingerprint of up to hundreds of mutations is a promising approach to identifying MRD in patients with breast cancer, but leveraging the power of this approach requires identifying enough tumor mutations to track in blood cfDNA [73]. When tumor tissue is not available, plasma ctDNA (pre-surgery) can be sequenced by NGS using a panel of 200–500 cancer genes, and mutated genes can be followed in plasma ctDNA post-surgery using a small panel that includes those genes (Figure 3B). Panels with a small number of genes are designed for target detection rather than to reveal emerging sub-clones.

## 5. Clonal Hematopoiesis of Indeterminate Potential

In addition to its low concentration, another challenge of using ctDNA for tumor genetic variants detection is that those variants can be confounded by clonal hematopoiesis of indeterminate potential (CHIP) [74]. Clonal hematopoiesis (CH) is an age-related condition defined by the abnormal expansion of clonally-derived hematopoietic stem cells carrying somatic mutations in leukemia-associated genes (especially *DNMT3A*, *TET2*, and *ASXL1*). CHIP is prevalent in the cfDNA of both healthy individuals and cancer patients, and its prevalence arises with aging (starts peaking at age 60 to 65) and certain therapeutic or environmental exposures [75]. The prevalence of CHIP is reported to be 20% to 95% in healthy adults aged 60–70 years, typically at a VAF < 0.1% [75,76]. In a study of somatic mutations in the cfDNA of 259 healthy individuals, 60% of the subjects had CHIP mutations [77]. The prevalence of CH increases in patients with hematologic as well as solid tumors. In cancer patients, the high rate of CHIP may also be influenced by prior exposure to chemotherapy [78,79].

CHIP may be a potential source of false positives when using ctDNA for tumor genetic variant detection due to the detection of non-reference variants in the blood plasma, which is especially problematic when the ctDNA mutant allele fraction is low in the setting of MRD detection [74]. Clonal hematopoiesis is associated with an increased risk for leukemias, cardiovascular disease, and mortality [28]. CH increases the risk of developing myeloid cancers by 10-fold, especially as myelodysplastic syndromes (MDS) and acute myeloid leukemia (AML) and bone marrow failure. CH increases the risk (2-fold) of cardiovascular disease and cardiovascular mortality. CH can be detected in cfDNA and also tumor tissues biopsies [80].

Although the number of CHIP variants in an individual was found to be associated with age, it is highly variable, with many young patients also having high levels of CHIP. Razavi et al. [28] studied 124 patients with metastatic cancer and 47 non-cancer controls. The cfDNA and WBC were studied using a panel of 410 genes and sequenced to a minimum targeted depth of 60,000×. An overall strong association of CHIP with age was found, as was an increased number of mutations by age, though there was extensive variability in the rate of CHIP among healthy controls and cancer patients in each category [28]. Some of the younger patients had high allele frequencies of the clonal hematopoiesis in their WBCs [28].

Most of the WBC matched mutations involved the canonical hematopoiesis genes, such as *DNMT3A*, *TET2*, *ASXL1*, *JAK2*, *SF3B1*, *PPM1D*, *U2AF1*, *MPL*, *IDH2*, *CBL*, *MYD88*; these genes are implicated in hematological cancers but are not commonly involved in solid tumor malignancy [23,74,75,80,81,82]. However, *TP53* and *KRAS* mutations are among the next most commonly mutated gene in CHIP [74,75,83], which is challenging given their high prevalence as driver mutations in solid tumors [23,84]. Tarazona et al. [23] reported two colon cancer patients without evidence of relapse who harbored CHIP mutations in *TP53* and *KRAS*. In renal-cell carcinoma (RCC), CHIP was found to affect cfDNA results in 43% of patients [85]. Jensen et al. studied genetic variants in prostate cancer patients (*n* = 69) and found CHIP variants in *ATM* (*n* = 5), *BRCA2* (*n* = 1), and *CHEK2* (*n* = 1)—all DNA repair genes used to determine poly (ADP) ribose polymerase inhibitors (PARPi) candidacy [86]. CHIP interference causing false-positive ctDNA biomarker assessments may result in patient harm from inappropriate treatment [86]. Thus, CHIP must be accounted for by sequencing matched WBCs to a similar depth [74]. Assays that involve sequencing of paired white blood cells (WBC) may lead to more specific ctDNA detection by enabling filtration of variants present in both ctDNA and WBC [22].

Finally, somatic clonal expansion and somatic mosaicisms are other sources of mutation that interfere with genetic variants from ctDNA detection. Recent work showed clonal expansion in normal tissue, including mutations coming from sun-exposed skin or the lungs or esophagus [87]. These other sources of interference should be kept in mind.

## 6. NGS-Based Commercially Available ctDNA Assays

Guardant360 (Guardant Health Inc., Redwood City, CA, USA) and Foundation One Liquid (Foundation Medicine, Cambridge, MA, USA) are cfDNA-based comprehensive commercial genomic profiling assays used to detect genetic alterations using plasma. Guardant360 (Guardant Health Inc., Redwood City, CA, USA) utilizes a 150 kb panel encompassing 73 cancer-related genes for hybrid capture, followed by NGS with an average sequencing depth of ~15,000×, noise filtering and molecular tracking, and variant calling for single nucleotide variants (SNVs), small insertions or deletions (indels), copy number alterations (CNAs), and fusions [88,89]. Validation of the Guardant360 assay in adult patients with advanced-stage solid tumors revealed high clinical sensitivity (85.9%) and SNV detection specificity of 97% [89]. Another commercial assay that utilizes hybrid capture-based NGS technology is Foundation One Liquid CDx (Foundation Medicine Inc., Cambridge, MA, USA) which includes 75 genes in its capture panel. In both commercial assays, genes with therapeutic, diagnostic, and prognostic relevance, as well as biomarkers that may serve to guide cancer treatment, were included. In addition, baited regions were included for the confident determination of the tumor mutation burden (TMB) and microsatellite instability (MSI) status, biomarkers associated with the prediction of response to immunotherapy. By not incorporating sequencing information from matched WBC in these commercial assays, their reported results may be confounded by CHIP. Torga et al. demonstrated very low congruence for the same patient-paired samples between these two assays [90].

Signatera (Natera Inc., Austin, TX, USA) is the first ctDNA assay optimized to detect MRD and assess treatment response for patients previously diagnosed with cancer [52,91,92,93]. The limit of detection for Signatera is 0.001% VAF which is equivalent to one mutant haploid genome in a background of 10,000 normal haploid genomes. Somatic variants are identified by the whole-exome sequencing (WES) of the primary tumor and the matched normal (whole blood) samples, and 16 somatic variants are chosen for each patient to follow them using plasma ctDNA (Figure 3A). This list of variants is then used to design PCR amplicons based on optimized design parameters. Multiplexed targeted PCR is conducted, followed by amplicon deep sequencing on the Illumina platform. Focusing on patient-specific variants enables ultra-deep sequencing (100,000× average depth of coverage) of each target. The resulting tumor signature, individualized to each patient’s tumor, is monitored throughout the patient’s disease course to detect the presence of ctDNA in the plasma [91,92,93]. Samples are considered ctDNA positive when ≥2 out of the 16 selected target mutations are present above a predefined threshold [94]. Signatera is validated across multiple cancer types to detect MRD up to 2 years earlier than standard diagnostic tools [52,91,92,93]; it detects CRC recurrence up to 16.5 months in advance of radiologic imaging [91] and early stage breast cancer up to two years earlier than imaging [93]. In a phase II clinical trial (NCT02644369), Signatera was used to evaluate tumor response to pembrolizumab in different types of advanced solid tumors (triple-negative breast cancer, squamous cell cancer of head and neck, high grade serous ovarian cancer, malignant melanoma and mixed solid tumors) [94]. ctDNA assays were performed using plasma samples obtained at baseline and every three cycles. Baseline ctDNA concentration correlated with progression-free survival (PFS), OS, clinical response, and clinical benefit [94]. In this study, all 12 patients with ctDNA clearance during treatment were alive with a median 25 month follow up [94]. This study demonstrates the potential clinical utility of ctDNA for surveillance in patients treated with pembrolizumab [94]. Signatera was approved by the Food and Drug Administration (FDA) for stage II and III CRC patients for two intended uses: (1) patient risk stratification after surgical resection, to inform adjuvant treatment decisions, and (2) recurrence monitoring with the same frequency as carcinoembryonic antigen (CEA), in patients with a previous cancer diagnosis but no ongoing evidence of disease.

## 7. Role of ctDNA in Management of Colorectal Cancer

CRC is the third most common cancer worldwide and the second leading cause of cancer-related death, with 1.9 million newly diagnosed cases each year [95]. In the US, the 5-year survival rate for people with localized stage CRC is 90%; if the cancer has spread to the surrounding organs and/or the regional lymph nodes, the 5-year survival rate is 72%; and for stage IV CRC, the 5-year survival rate is 14% [96]. The current standard of care for patients with early stage CRC includes the surgical resection of the tumor followed by adjuvant chemotherapy in selected patients [97,98]. Most patients with stage II CRC are not treated with chemotherapy; however, approximately 10–15% have residual disease after surgery [99]. The identification of this patient population and treatment with chemotherapy could potentially reduce their risk of recurrence. Conversely, most patients with stage III CRC receive chemotherapy despite more than 50% being cured by surgery [100,101,102]. Furthermore, approximately 30% of the chemotherapy-treated patients with stage III CRC experience recurrence, making them candidates for additional therapy [99]. Thus, improved tools to identify the patient population who would benefit from chemotherapy are greatly needed.

Early diagnosis of recurrent disease is another significant unmet clinical need in CRC. After the completion of definitive treatment, surveillance is recommended to detect recurrence sufficiently early for potentially curative surgery [97,98]. Despite surveillance, many recurrence events are detected late, and only 10–20% of metachronous metastases are treated with curative intent [103]. Nevertheless, currently recommended follow-up programs, consisting of the use of imaging techniques plus plasma carcinoembryonic antigen (CEA) monitoring, are suboptimal, failing to detect MRD and mostly diagnosing only far advanced relapses [104]. The current goal standard for assessing initial disease bulk and for defining treatment response is the image-based Response Evaluation Criteria in Solid Tumors (RECIST). RECIST limitations include low inter- and intra-observer reproducibility and limited categorization [105] and CEA’s lack of sensitivity and specificity [106,107,108]. Therefore, there is a need for better biomarkers that can detect patients at high risk of recurrence to enable appropriate follow-up and therapeutic strategies for early recurrence detection and curative treatment [109].

Plasma ctDNA has emerged as a promising biomarker for the longitudinal assessment of tumors throughout disease management. In CRC, there are multiple indications for which ctDNA can assist with clinical decision making. Recently, the Colon and Rectal-Anal Task Forces of the United States National Cancer Institute provided detailed guidance in the standardization and efficient development of ctDNA technology [51]. In the setting of metastatic CRC, they recommended that the ideal ctDNA assay should involve a multigene panel that enables high-depth sequencing of the most commonly altered genes in order to capture the changes associated with non-targeted as well as targeted therapies. With such an assay, the presence of any CRC-related somatic alteration could be used to indicate a positive test, and the highest VAF of the alteration could be used to define ctDNA concentration [51]. The assay would need to be performed prior to the start of the treatment and then again soon after starting treatment to guide the determination of an early clinical response [51]. The sequencing of the DNA from CRC identified several genes that are recurrently mutated [110], and these tumor-specific mutations can be detected in the cfDNA of peripheral blood in most patients with metastatic disease. Genetic variants from 1397 patients with advanced CRC were studied using ctDNA and compared with data from three independent tissue-based CRC sequencing databases [111]. The spectrum and frequency of genomic alterations identified in ctDNA demonstrate a striking similarity to results from these three large CRC tumor tissue sequencing databases [111]. The genes most mutated in colon cancer patients are *KRAS*, *BRAF*, *PIK3CA*, *TP53*, *APC*, *FBXW7*, *NRAS*, *CTNNB1*, *SMAD4*, *PTEN*, *ERBB3*, and *EGFR* [23,112].

Epigenetic analysis of cfDNA/ctDNA might contribute to the identification of gene hypermethylation [113,114]. The methylation of *HLTF* and *HPP1* genes was associated with worse survival in CRC [115,116]. Lee et al. [117] analyzed the promoter methylation of the Septin 9 gene among patients with stage I–II CRC and suggested that its methylation might be associated with lower disease-free survival. Herbst et al. [118] suggested that the detection of *HPP1* methylation in cfDNA might be used as a prognostic marker and an early marker to identify patients who will likely benefit from a combination of chemotherapy and bevacizumab. However, the methylation status of cfDNA and its application to detect MRD and response to treatment are studied less intensively [119,120].

## 8. Role of ctDNA in the Detection of Minimal Residual Disease in Colorectal Cancer

In patients with stage II colon cancer (~25% of all colorectal cancer), management after surgical resection remains a clinical dilemma, with about 80% cured by surgery alone. Adjuvant chemotherapy is more frequently offered to high-risk stage II patients. However, an overall survival benefit from adjuvant therapy in patients with stage II colon cancer, including those with the high-risk disease based on standard clinicopathologic criteria or gene signatures, remains to be conclusively demonstrated [121]. The challenge in demonstrating a benefit is in part due to the overall low risk of recurrence in patients with stage II colon cancer. The decision to treat or not to treat stage II colon cancer patients with adjuvant chemotherapy remains one of the most challenging areas in colorectal oncology. Currently, up to 40% of stage II patients undergo adjuvant therapy in routine clinical care [122], committing to 6 months of chemotherapy, with the associated risk of potentially serious adverse events and without a method to monitor the impact of adjuvant therapy, for an absolute risk reduction of 3–5%. Although multiple clinicopathological markers are now validated and can be combined to define low- and high-risk groups, only a minority of defined high-risk patients will develop recurrence. Diagnostic approaches that better predict the disease course in this patient population are therefore urgently required.

Tie et al., using a tumor-informed Safe-SeqS platform-based ctDNA assay, reported two prospective, multicenter cohort studies, one in stage II (*n* = 230) [21] and the other in stage III (*n* = 96) patients [123], showing that ctDNA significantly outperformed standard clinicopathologic characteristics as a prognostic marker. In these studies, tumor tissue was analyzed for somatic mutations in 15 genes commonly known to be mutated in CRC, and one mutation identified in the tumor tissue (the mutation with the highest mutant allele fraction, MAF) was selected for ctDNA testing in each patient. Among 230 patients with resected stage II colon cancer, the presence of ctDNA in postoperative plasma samples was strongly associated with recurrence in those who did not receive adjuvant chemotherapy (Figure 4A) [21]. Among the patients in the stage II cohort [21] who did not receive adjuvant chemotherapy (*n* = 178), 79% of the patients (11 out of 14) with detectable ctDNA postoperatively (4 to 10 weeks after surgery) had cancer recurrence at a median follow up duration of 27 months (HR 18, 95% CI 7.9–40; *p* < 0.001) (Figure 4A). Conversely, only 9.8% (16 out of 164) of the patients with undetectable postoperative ctDNA had a cancer recurrence (Figure 4A) [21]. In conclusion, Tie et al. [21] demonstrated that stage II colon cancer patients who were ctDNA positive postoperatively were at extremely high risk of radiologic recurrence when not treated with chemotherapy. This risk is greater than in patients with stage III colorectal cancer, who are routinely treated with adjuvant therapy. Conversely, patients with negative ctDNA postoperatively were at low risk of radiologic recurrence (3-year RFS of 90%), not dissimilar to patients with stage I colorectal cancer [124], defining a group where adjuvant chemotherapy is less likely to be helpful [21]. Kaplan–Meier estimates of recurrence-free survival (RFS) at 3 years were 0% for the ctDNA-positive and 90% for the ctDNA-negative groups (Figure 4A). These studies showed that ctDNA could be used to monitor MRD in early stage CRC patients.

Tie et al. [123] also studied ctDNA in stage III colon cancer patients. In this study, ctDNA was detectable in 20 out of 96 (21%) patients postoperatively (4–10 weeks after surgery), and the recurrence-free survival (RFS) at 3 years in this group was 47% (95% CI, 24–68%) compared to 76% in those with undetectable postoperative ctDNA (95% CI, 61–86%) (Figure 4B) [123]. Patients with detectable ctDNA after surgery had an increased risk of recurrence (HR, 3.8; 95% CI, 2.4–21.0; *p* < 0.001) (Figure 4B). Like stage II patients, postoperative ctDNA status remained independently associated with recurrence-free interval (RFI) after adjusting for known clinicopathologic risk factors (HR, 7.5; 95% CI, 3.5–16.1; *p* < 0.001) [123]. In conclusion, in patients with stage II and III colon cancer, ctDNA may be a useful prognostic marker after surgery and could guide initial adjuvant treatment. In locally advanced rectal cancer, postoperative ctDNA detection was also predictive of recurrence [125].

Tie et al. [126], using a meta-analysis approach from their three studies [21,123,125], showed that patients with non-metastatic CRC with detectable ctDNA after surgery had poorer 5-year recurrence-free (38.6% vs. 85.5%; *p* < 0.001) and overall survival (64.6% vs. 89.4%; *p* < 0.001). Analysis of ctDNA 4 to 10 weeks after surgery is a powerful prognostic marker [126]. In a study by Reinert et al. [91], conducted in a cohort of 125 CRC patients (stages I to III), ctDNA was quantified in the preoperative and postoperative plasma samples. The study showed that the patients with detectable ctDNA at postoperative day 30 were seven times (HR, 7.2; 95% CI, 2.7–19.0; *p* ≤ 0.001) more likely to have cancer recurrence compared to those with undetectable ctDNA [91]. ctDNA status was the only significant prognostic factor associated with recurrence-free survival (RFS) [91]. Other studies in CRC showed similar results [23]. In conclusion, ctDNA could be used to monitor MRD in early stage CRC patients postoperative. In other early stage cancers, such as breast [38] and pancreatic cancer [53], it was also shown that ctDNA detection after curative surgery (postoperative) was predictive of early cancer relapse. Therefore, ctDNA is a robust predictor of disease recurrence, as shown by the studies in CRC and other types of cancers.

## 9. Role of ctDNA in Assessing the Efficacy of Adjuvant Therapy in Colorectal Cancer

Tie et al. also demonstrated in stage II colon cancer patients that being ctDNA-positive at the completion of adjuvant chemotherapy treatment predicted a very high risk of radiologic recurrence [21]. ctDNA detection immediately after completion of chemotherapy was associated with poorer RFS (HR,11; 95% CI, 1.8 to 68; *p* ≤ 0.001) (Figure 5A). In this study, all patients had recurrence if ctDNA was detectable after chemotherapy. The median lead time from ctDNA detection to radiologic recurrence was over 5 months, which might be sufficient to change patient management. Personalized serial measurements of ctDNA during adjuvant therapy could be a real-time marker of adjuvant therapy impact. In stage III patients, Tie et al. [123] reported that the ctDNA status of the post-chemotherapy sample was strongly associated with recurrence-free interval (RFI) (HR, 6.8; 95% CI, 11.0–157.0; *p* < 0.001) [123]. The three-year recurrence-free survival (RFS) was 30% (95% CI, 9–55%) for cases with detectable ctDNA and 77% (95% CI, 60–87%) for those with undetectable ctDNA after chemotherapy (Figure 5B).

In patients with stage III CRC, a positive postsurgical ctDNA finding and a positive ctDNA finding after chemotherapy (“Positive-positive”; Figure 6A), an inferior RFI was seen compared to patients in whom ctDNA became undetectable after chemotherapy (Positive-negative, Figure 6A) (HR, 3.7; 95% CI, 1.1–17.0; *p* = 0.04) [123]. In patients with a negative postsurgical ctDNA finding and a negative ctDNA results after chemotherapy (“Negative-negative”; Figure 6B), there was a superior RFI compared to patients in whom ctDNA became detectable after chemotherapy (Negative-positive; Figure 6B) (HR, 6,5; 95% CI, 7.2–642.0, *p* < 0.001) [123].

In another study, Reinert et al. [91] reported a 17 times higher risk of recurrence in patients with CRC if ctDNA was detectable after completion of chemotherapy (HR, 17.5; 95% CI, 5.4–56.5; *p* < 0.001) [91]. In this study, 3 out of 10 patients (30%) with detectable postoperative ctDNA cleared ctDNA after chemotherapy and were disease-free long term, and the other 7 patients with detectable ctDNA after chemotherapy had disease relapse [91]. During surveillance after definitive therapy, ctDNA-positive patients were more than 40 times more likely to experience disease recurrence than ctDNA-negative patients (HR, 43.5; 95% CI, 9.8–193.5; *p* < 0.001) [91]. Serial ctDNA analyses revealed disease recurrence up to 16.6 months ahead of standard-of-care radiologic imaging (mean, 8.7 months; range, 0.8–16.5 months) [91].

Tarazona et al. [23] studied colon cancer patients and reported an 85.7% recurrence in patients with detectable ctDNA post-chemotherapy (HR 10.02; 95% CI 9.202–307.3; *p* < 0.0001). In this study, one out of seven patients cleared ctDNA after chemotherapy and remained disease-free in the long term [23]. Genetic variants were studied in tumor tissue using a custom-targeted NGS panel, and two variants with the highest VAF in each patient were selected to track ctDNA in the plasma samples by ddPCR [23]. The detection of ctDNA post-operative at follow-up after adjuvant chemotherapy in patients with localized colon cancer preceded radiological recurrence with a median lead time of 11.5 months [23].

All these studies provide evidence supporting the utility of ctDNA to inform clinicians on the efficacy of adjuvant therapy and suggest that clearance of ctDNA after chemotherapy could be considered a surrogate marker of survival and adjuvant therapy effectiveness. ctDNA may be used as a real-time marker of adjuvant therapy effectiveness, opening new strategies for enrolling high-risk patients with detectable ctDNA in different therapies. Owing to the high specificity of ctDNA for the prediction of disease recurrence, patients with detectable ctDNA might be considered candidates for the escalation of adjuvant therapy over the standard-of-care approach to reduce the risk of disease recurrence [51]. Conversely, patients who lack detectable ctDNA, as determined using a sufficiently sensitive assay, and who have a low risk of disease recurrence might benefit from de-escalation to less-intense adjuvant therapies that reduce the risk of toxicities [51]. ctDNA analysis can potentially change the postoperative management of CRC by enabling risk stratification, chemotherapy monitoring, and early relapse detection.

## 10. Potential Role of ctDNA in Surveillance for Colorectal Cancer Patients

Regardless of whether patients received adjuvant therapy, the early detection of recurrence during follow-up is associated with improved survival in patients with early stage CRC [109]. However, the biomarker now used as the standard of care for CEA has limited sensitivity and specificity [106,107,108]. CT scans improve the detection of recurrence but are associated with radiation exposure, high cost, inter-reader variability, and a high rate of false positivity [127]. Additionally, by the time CT detection occurs, it may be too late for surgical management. Several studies suggested that ctDNA can diagnose CRC recurrence much earlier than standard surveillance methods [91,125,128,129]. In these studies, detectable ctDNA during surveillance was associated with cancer relapse, and ctDNA detection preceded radiologic relapse by a median time interval from 3 to 11.5 months.

## 11. ctDNA in Monitoring Response to Treatment in Metastatic Colorectal Cancer

Radiographic imaging and serum CEA levels are currently used to monitor disease status in metastatic CRC (mCRC) patients. However, serum CEA levels might only be elevated in 70–80% of patients [107,130]. Several studies employed ctDNA as a biomarker of metastatic disease to monitor disease response to systemic therapy and to assess the overall disease burden. Early changes in ctDNA during treatment with standard therapies are shown to predict radiological responses in patients with mCRC [37,131,132]. A decrease in the ctDNA level during systemic therapy in mCRC correlates with tumor response [11,37,132,133,134]. Garlan et al. [132] showed in a study of mCRC patients that reductions in ctDNA concentration of ≥80% after first-line or second-line chemotherapy were associated with a significantly improved objective response rate (47.1% versus 0%; *p* = 0.003) and longer median PFS (8.5 months versus 2.4 months; HR 0.19, 95% CI 0.09–0.40; *p* < 0.0001) and OS (27.1 months versus 11.2 months; HR 0.25, 95% CI 0.11–0.57; *p* < 0.001) [132]. These authors studied changes in ctDNA, before and after chemotherapy, by the identification of somatic alterations or the hypermethylation of two genes (*WIF1* and *NPY*) and concluded that early change (after cycle one or cycle two) in ctDNA level was a marker of therapeutic efficacy [132].

Tie et al. evaluated 53 mCRC (treatment naïve) patients receiving standard first-line chemotherapy by monitoring ctDNA levels [37]. Tumors were sequenced using a panel of 15 genes frequently mutated in mCRC to identify candidate mutations for ctDNA analysis. For each patient, one tumor mutation was selected to assess the presence and the level of ctDNA in plasma samples using Safe-SeqS. Results indicated that patients with a reduction in ctDNA just before cycle two also had a radiological-confirmed response 8–10 weeks later [37]. Significant reductions in ctDNA (median 5.7-fold; *p* < 0.001) levels were observed before cycle two, which correlated with CT responses at 8–10 weeks (odds ratio = 5.25 with a 10-fold ctDNA reduction; *p* = 0.016). Overall, 14/19 (74%) patients who had a ≥10-fold reduction in ctDNA levels had a radiologic response measure at 8–10 weeks, while only 8/23 (35%) patients with lesser reduction in ctDNA levels responded [odds ratio = 5.25; 95% CI 1.38–19.93; *p* = 0.016]. Major reductions (>10-fold) versus lesser reduction in ctDNA pre-cycle two were associated with a trend for increased PFS (median 14.7 versus 8.1 months; HR = 1.87; *p* = 0.266). The optimal criterion for predicting response to therapy was ≥10-fold change in ctDNA after cycle one of chemotherapy. Patients who met this criterion experienced a trend of longer PFS than patients with <10-fold drop in ctDNA (median PFS, 14.7 versus 8.1 months; HR = 1.87; 95% CI 0.62–5.61). In metastatic CRC, data suggest that ctDNA changes can occur rapidly in response to systemic therapy, with ctDNA variations at 2 weeks being predictive of subsequent radiographic results in restaging studies at 2 months [37]. In conclusion, early changes in ctDNA in treatment naïve mCRC patients during first-line chemotherapy predict the later radiologic response [37]. No significant relationship was found between fold change and OS [37]. ctDNA might be incorporated as a biomarker to assess mCRC patient response to treatment. If a patient’s non-response to each treatment could be reliably assessed earlier, such as with serial ctDNA analysis, an earlier switch to an alternative therapy may be of benefit, minimizing the side-effects of the ineffective therapy and providing the opportunity for a more effective one [37]. Another important impact of serial ctDNA measurements would be in patients with the non-measurable disease by RECIST criteria, where a reliable measure of ctDNA response would assist clinical decision making [37].

In conclusion, the results of several studies in CRC consistently demonstrate the potential use of ctDNA as a prognostic tool. The findings of these studies showing the role of ctDNA as a biomarker to detect MRD, follow response to treatment, and surveille disease are summarized in Table 1.

## 12. Using ctDNA to Detect Secondary Drug Resistance in Metastatic CRC

Tumor genotype plays an important part in determining drug resistance in patients with mCRC. Genotyping of tumor tissue can help with the selection of patients with tumor amenable to treatment; however, the value of testing a tumor sample is limited by inter- and intra-tumor heterogeneity. Moreover, archival tissue will not show the genotypic changes that occurred since the sample was obtained. However, since tissue-based sequencing compendia rely on early stage and treatment-naïve tumors, these databases have limited insights into acquired resistance mutations. Large ctDNA cohorts can more readily provide non-invasive access to patients with advanced tumors and may offer unique insight into resistance mechanisms emerging under the selective pressure of systemic therapies. The standard treatment for mCRC includes drugs that target the molecular drivers of colorectal cancer pathogenesis, such as VEGF and EGFR [135,136]. *EGFR* mutations in the extracellular domain (ECD: amino acids 334 to 505) mediate resistance by blocking the binding of anti-EFGR antibodies (e.g., cetuximab, panitumumab). The potential for EGFR ECD mutations to drive resistance to anti-EGFR antibodies is documented through ctDNA and tumor biopsies. However, studies using ctDNA identified a novel cluster of mutations (cluster 1) in EGFR ECD. This previously unreported cluster of EGFR ECD mutations involving V441 and S442 accounted for 25% of all EGFR EDC mutations [111].

The use of anti-EGFR antibody treatment is restricted to patients with *KRAS* wild-type tumors, some of whom acquire *KRAS* mutations during treatment as a mechanism of drug resistance [137,138]. In 2012, two independent groups uncovered *KRAS* alterations as mechanisms of emerging resistance to anti-EGFR therapy in CRC with ctDNA analysis [137,138]. These studies revealed that ctDNA anticipated the emergence of *KRAS* resistant subclones 10 months before radiographic progression. Other studies on ctDNA showed that mutated *KRAS* alleles detected at progression start declining when the anti-EGFR blockage is withdrawn, suggesting a potential role for anti-EGFR re-challenge [139,140]. The withdrawal of EGFR blockade drugs after progression leads to further clonal evolution that can be exploited pharmacologically.

## 13. Clinical Trials Using ctDNA as Biomarker in CRC

Clinical trials are already evaluating the potential value of ctDNA in the early stage CRC setting to select patients that could benefit from further treatments. The Australian trial DYNAMIC-II (ACTRN12615000381583) will determine the effect of the use of ctDNA to guide adjuvant chemotherapy in stage II CRC patients. DYNAMIC-III (ACTRN12617001566325) is a phase II/III study enrolling stage III colon cancer patients who will be randomized to be treated according to post-operative ctDNA results (Arm B: ctDNA-informed) or as per the standard of care (Arm A: SOC). Patients who are ctDNA-negative will be managed with a de-escalation adjuvant treatment, and those who are ctDNA-positive will be managed with an escalation adjuvant treatment strategy. In the standard of care arm, patients will receive adjuvant chemotherapy as per the standard of care.

The COBRA study (NCT-04068103) is a phase II/III study of ctDNA as a predictive biomarker in adjuvant chemotherapy in patients with stage IIA colon cancer; this study will identify patients with colon cancer after surgery who will benefit and those who will not benefit, from receiving chemotherapy. The IMPROVE-IT trial (NCT03748680) is opened for patients with surgically removed adenocarcinoma of the colon or rectum with pathologically stage I or II diseases, and radical resection with detectable ctDNA in two weeks postoperative plasma sample, and with no indication for adjuvant chemotherapy according to Danish Colorectal Cancer Group (DCCG) guidelines. The primary aim of the study is to investigate if the use of standard adjuvant chemotherapy improves the disease-free survival in patients with MRD detected by ctDNA, where adjuvant chemotherapy is not standard treatment.

The Netherlands trial MEDOCC-CrEATE (NL6281/NTR6455) investigates whether adjuvant chemotherapy (ACT) reduces the risk of recurrence in stage II colon cancer patients with detectable ctDNA after surgery [141].

CIRCULATE-Japan is a project that aims to detect MRD and measure treatment responsiveness in resectable CRC using ctDNA testing [142]. CIRCULATE-Japan encompasses both “de-escalation” and “escalation” trials for ctDNA negative and positive patients, respectively, to answer whether measuring ctDNA postoperative has prognostic and/or predictive value [142]. It is composed of one observational study (GALAXY) and two randomized phase III trials, the VEGA and ALTAIR trials [142]. The GALAXY study is a prospectively conducted large-scale registry designed to monitor ctDNA for patients with stage II to IV or recurrent CRC who undergo complete surgical resection. A total of 2500 patients will be enrolled, and CT imaging will be performed every 6 months after surgery for 7 years [142]. The VEGA trial is designed for ctDNA negative patients at 4 weeks after curative surgery in the GALAXY study with high-risk stage II or low-risk stage III colon cancer to test whether postoperative surgery alone is non-inferior to the standard therapy of capecitabine plus oxaliplatin [142]. Finally, the ALTAIR trial is for patients with resected CRC who are ctDNA positive in the GALAXY study and is designed to establish the superiority of trifluridine/tipiracil as compared to placebo [142]. These ctDNA guided adaptive platform trials will accelerate clinical development toward further precision oncology in the field of adjuvant therapy.

## 14. Conclusions

The emerging field of ctDNA research has opened new avenues for cancer diagnostics over the past 5 years, with important clinical opportunities for personalized medicine in oncology. In patients with stages II and III CRC, it was demonstrated that ctDNA may be a useful prognostic marker after surgery and could guide initial adjuvant treatment and could be useful to monitor recurrence. ctDNA analysis can potentially change the postoperative management of CRC by enabling risk stratification, chemotherapy monitoring, and early relapse detection. Several studies in CRC suggested that ctDNA can be used in surveillance and detects recurrence 3–11.5 months earlier than imaging. In the CRC metastatic setting, decreases in ctDNA level during systemic therapy (first or second line of therapy) correlate with tumor response. The clinical use of ctDNA as a biomarker in cancer care will depend on the standardization of pre-analytic and analytic procedures [143]. There are several points that must be addressed before cfDNA/ctDNA enters the realm of clinical practice, including the agreement on sample-collection methods, methods for cfDNA isolation and quantification, the methodology to identify genomic alterations, and the use of NGS-based gene panels. Finally, for ctDNA assays to enter clinical practice, it is necessary to prove the clinical utility of ctDNA, and this can only be achieved in interventional clinical trials where the biomarker results determine the treatment choice [143]. The utility of ctDNA to support patient selection for early phase clinical trials is currently being investigated, and the results from new and ongoing trials will be paramount to this use of ctDNA.

## Figures and Tables

**Figure 1 ijms-23-04441-f001:**
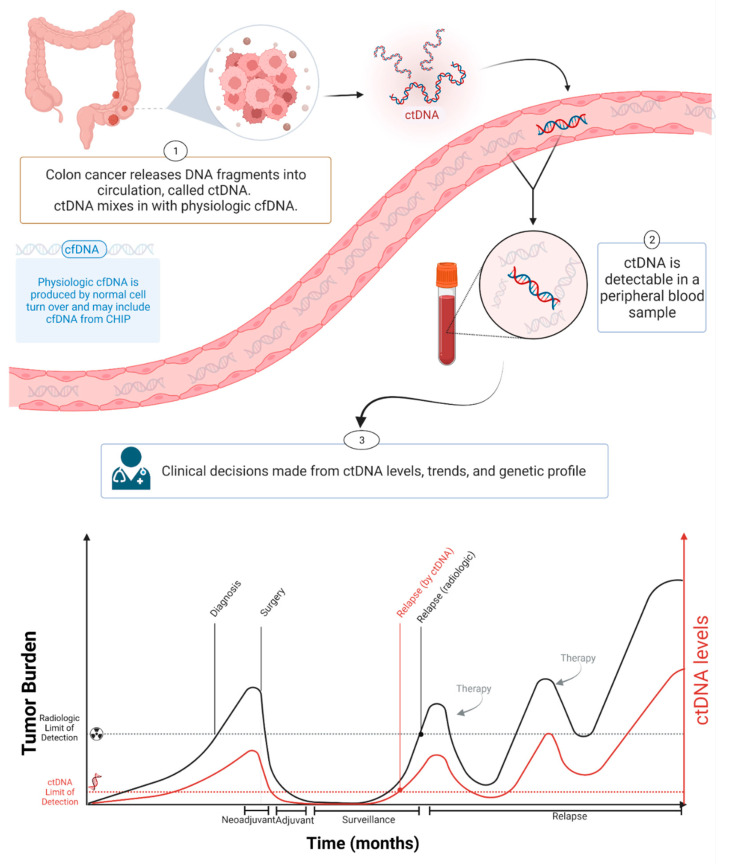
ctDNA during cancer progression. Created with BioRender.com.

**Figure 2 ijms-23-04441-f002:**
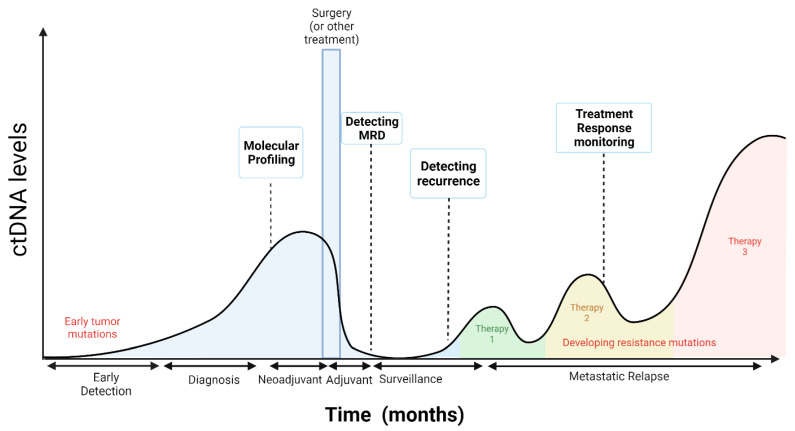
Potential applications of ctDNA detection assays in early and late stages of solid tumors. Adapted from Wan et al. [25]. Created with BioRender.com.

**Figure 3 ijms-23-04441-f003:**
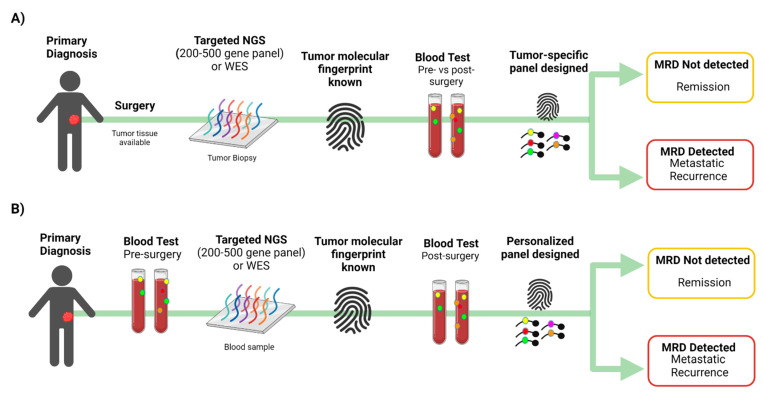
The number of tracked mutations impacts the ctDNA detection rate. Overview of MRD testing, including why tracking many mutations is important, particularly when there is a low fraction of cancerous cfDNA in the bloodstream. (**A**) Tumor tissue is available; (**B**) Tumor tissue is not available. WES, whole-exome sequencing. Created with BioRender.com.

**Figure 4 ijms-23-04441-f004:**
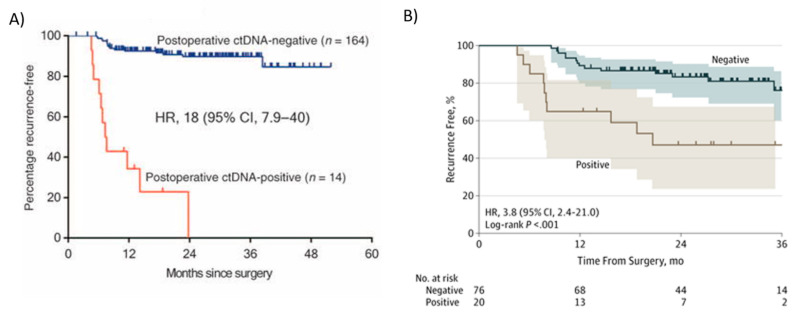
Kaplan–Meier estimates of recurrence-free interval according to ctDNA status in patients with stage II and III colon cancer after surgery (postoperative). In (**A**) Recurrence-free survival (RFS) in colorectal stage II patients post-surgery not treated with adjuvant chemotherapy. Patients with ctDNA-positive status postoperative had a markedly reduced RFS compared with those with a ctDNA-negative status. From Tie et al. [21]; (**B**) Kaplan–Meier estimates of recurrence-free interval according to ctDNA status in patients with stage III colon cancer after surgery. From Tie et al. [123].

**Figure 5 ijms-23-04441-f005:**
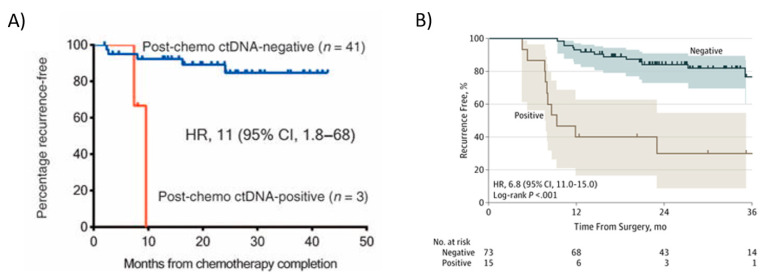
Kaplan–Meier estimates of recurrence-free interval according to ctDNA in colon cancer patients after adjuvant chemotherapy. ctDNA status after adjuvant chemotherapy in patients with colon cancer. (**A**) Stage II; ctDNA positivity immediately after completion of chemotherapy was associated with poorer RFS. From Tie et al., 2016 [21]. (**B**) In Stage III. From Tie et al. [123].

**Figure 6 ijms-23-04441-f006:**
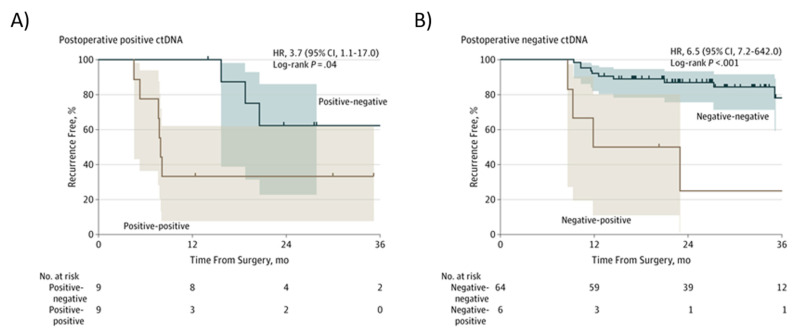
Kaplan–Meier estimates of recurrence-free interval according to ctDNA in stage III colon cancer patients after chemotherapy. (**A**) Postoperative positive ctDNA; (**B**) Postoperative negative ctDNA. From Tie et al. [123].

**Table 1 ijms-23-04441-t001:** ctDNA as a biomarker of MRD, therapeutic efficacy, and surveillance in CRC patients. Studies of plasma ctDNA in CRC and their main findings are shown. The number of patients included in each study is indicated (*n*). PFS, progression-free survival; OS, overall survival; CT, computed tomography.

Study	Tumor Type and Stage	Findings
Tie et al. [21]	Colon cancer Stage II (*n* = 230)	Patients ctDNA positive after curative intent surgery are at high risk of recurrence (HR, 28). Patients ctDNA positive at completion of adjuvant chemotherapy are at high risk of recurrence (HR,11; *p* ≤ 0.001).
Tie et al. [123]	Colon cancer Stage III (*n*= 96)	Patients ctDNA positive after surgery have poor outcomes despite adjuvant chemotherapy (3 years RFI 47% vs. 76% in those with ctDNA negative post-surgery) (HR, 3.8; *p* <0.001). When ctDNA is detectable despite adjuvant chemotherapy, the risk of recurrence is higher than when ctDNA is undetectable after treatment (HR, 6.8; *p* <0.001).
Tie et al. [125]	Rectal cancer Locally advanced (*n* = 159)	After surgery, 11 of 19 (58%) patients with ctDNA positive and 12 of 140 (8.6%) with ctDNA negative had recurrence (HR, 13; *p* < 0.001). Postoperative ctDNA detection was predictive of recurrence irrespective of adjuvant chemotherapy use (with chemo.: HR,10.0; *p* < 0.001; without chemo.: HR, 22.0; *p* < 0.001).
Tie et al. [126]	CRC Stage II–III	Meta-analysis; Studies from references: 21, 109, and 111 (*n* = 485).
Reinert et al. [91]	CRC Stage I–III (*n* = 130)	ctDNA positive patients at day 30 postoperatively were 7 times more likely to have recurrence compared to ctDNA negative patients (HR, 7.2; *p* < 0.001). ctDNA positive patients shortly after completion of chemotherapy had 17 times higher risk of recurrence compared with ctDNA negative ones (HR, 17.5; *p* < 0.001). During surveillance, ctDNA positive patients were more than 40 times more likely to have recurrence than ctDNA negative patients (HR, 43.5; *p* < 0.001).
Tarazona et al. [23]	Colon cancer Stage I–III (*n* = 150)	Detection of ctDNA after surgery and in plasma samples during follow up were associated with poorer disease-free survival (HR, 17.56; *p*= 0.0014 and HR, 11.33; *p* = 0.0001, respectively). ctDNA positive patients after adjuvant chemotherapy were at high risk of recurrence compared with ctDNA negative ones (HR, 10.02; *p* < 0.0001).
Scholer at al. [128]	CRC Stages I–IV (*n* = 45)	Patients with localized disease (Stages I-III) treated with curative intent and who were ctDNA positive within the first postoperative trimester had a high risk (100%) of relapse (HR, 37.7; *p* < 0.001); patients who were ctDNA negative were at a low risk of relapse (3-year RFS of 75%). Stage IV patients with liver metastasis treated with curative intent who were ctDNA positive within the first postoperative trimester were at high risk of relapse (HR, 4.9; *p* = 0.007).
Wang et al. [129]	CRC Stage I–III (*n* = 58)	Patients ctDNA positive postoperatively had 77% (10 of 13 patients) recurrence versus 0% (0 of 45 patients) with negative ctDNA. Patients who remained ctDNA negative through follow up had no relapse. Patients with ctDNA positive postoperatively could still be cured by chemotherapy.
Garlan et al. [132]	CRC Stage IV (*n* = 82)	Early change in ctDNA level (after cycle 1 or 2) was a marker of therapeutic efficacy in mCRC treated with first- or second line chemotherapy alone or in combination with targeted therapy.
Tie et al. [37]	CRC IV Chem. naive (*n* = 53)	Significant reduction in ctDNA (median 5.7-fold; *p* < 0.001) levels were observed before cycle 2 during first-line chemotherapy, which correlated with CT responses at 8–10 weeks. Major reductions (>10-fold) versus lesser reduction in ctDNA pre-cycle 2 chemotherapy were associated with a trend for increased PFS (median 14.7 versus 8.1 months).

## Data Availability

Not applicable.

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
