# Peer review of "Circulating Tumor DNA in Precision Oncology and Its Applications in Colorectal Cancer"

_ijms, 2022, doi:10.3390/ijms23084441_

Round 1

Reviewer 1 Report

This review article written by Arisi et al provides a succinct overview of ctDNA and its applications in oncology and particularly in CRC. The article touches on several major issues, such as the problem of clonal hematopoiesis and the wide variety of platforms available for ctDNA analysis, giving the reader a global view on the field. The focus on CRC is highly relevant given its high incidence and mortality rates. Presented in a clear, comprehensive and logical manner, the review describes the most important studies assessing the clinical utility of ctDNA in the management of CRC. The references provided are appropriate and adequate and  the figures, most of them reproduced from original studies, illustrate key findings that highlight the potential of ctDNA as a biomarker.

I only have some minor comments as stated below.

MINOR COMMENTS

  • It is rather unusual to have figures in the abstract and start a paper by presenting a figure. I would recommend omitting Figure 1 references from the abstract and incorporating them in Section “1. Cell-Free DNA (cfDNA) and Circulating Tumor DNA (ctDNA)”.
  • Line 21: “is lower that”. Please correct.
  • Line 41: “though”. Please correct.
  • Lines 204-8: It should be noted that BEAMing – esp. the format used for ctDNA detection- is also a type of digital PCR and this should be made clear by the authors in the manuscript (for example see the following references: https://academic.oup.com/clinchem/article/65/11/1405/5715869 and https://www.nature.com/articles/s41416-019-0457-y).

BEAMing combines emulsion PCR with flow cytometry, which makes it such a powerful tool, and this should also be added in the text. The authors should also briefly describe the basic principle of ddPCR.

  • Line 211: Both dPCR platforms (ddPCR and BEAMing) have this limitation- Please correct.
  • Line 224: “NGS can be applied to the targeted panel” do the authors mean “NGS can be applied using the targeted panel”?
  • Line 232: “would requires” Please correct.
  • Line 314: The authors describe in this section only NGS-based commercially available ctDNA assays, therefore the title should be changed accordingly (there are also several RT- and digital- PCR – based assays in the market).
  • Section 8 “ctDNA and MRD in CRC” would benefit from a table summarizing the studies and the clinical outcomes with the ctDNA detected.
  • One important study in the field is the “Genomic Landscape of Cell-Free DNA in Patients with Colorectal Cancer (Cancer Discov; 8(2); 164–73)” which provided one of the first examples of large-scale genomic profiling of cfDNA in CRC patients and identified novel EGFR ectodomain mutations, which the authors should also incorporate in their manuscript.

Reviewer 2 Report

The paper is an overview of tumor specific circulating DNA (ctDNA) in the treatment of malignant diseases with special focus on colorectal cancer.

This topic has been addressed in many recent reviews and the present paper adds no new information. It is based on published articles with presentation of previously published figures.

There is no relevant criticism of the literature, which in my opinion should be an essential part of a review paper.

The authors also fall short in their selection of literature and neglect discussion of conflicting results.

The authors complete neglect epigenetic changes, which is a major drawback when it comes to aberrant methylation. There is a considerable number of papers describing the importance of tumor specific methylated DNA (meth-ctDNA). One example is the French IDEA study, which was practice changing with respect to adjuvant chemotherapy in colon cancer. The analysis of meth-ctDNA proved its clinical importance in this context. I think a contemporary review must include this trial. The same applies to the large number of papers on meth-ctDNA in metastatic disease.

In conclusion, the overview has shortcomings in relation to selected literature. The authors completely neglect meth-ctDNA and thus show one side of the coin only. Consequently, I cannot recommend the paper for publication in a journal of a high scientific standard.

Author Response

Dear reviewer,
Thank you for taking your time to review our manuscript. I am very sorry that you have reached the conclusion that our review article is useless. It took me a long time to work in this review article in which I went through so many publications. I was really impressed by the work of several groups in particular the australian group of Tie et al.  It was a very rewarding experience for me that allow me to appreciate the work from the scientific community in this field that it is so important for cancer patients like myself. 

I focused on ctDNA assays that are near to be use in the clinical practice which is a field where I have my expertise. I am not an expert in ctDNA methylation.

I am guest editor of this special issue on "Liquid Biopsies in Oncology" for which I wrote my review article. I have invited several scientists to write on different
topics.  It will be a pleasure for me to invite you to write an article on ctDNA methylation that it is your area of expertise according to your comments.

Waiting for your email.

Sincerely yours,

Sandra Viviana Fernandez

[email protected]

Reviewer 3 Report

Dear authors,

I’ve read with great interest your review on such a hot topic.

Here my comments:

I suggest you to remove “(ctDNA)” from the title of the article and to remove “(figure 1)” from the abstract.

The use of figures taken from other articles, even if you obtained the permissions, in my honest opinion, is not a good option: in fact, I would expect them in a congress presentation rather than a review article. A possible solution would be to modify them by adopting the same style for all figures you have chosen, for example a simplified graphical style in greyscale.

Chapter 4: about PCR, you have not cited the quantitative PCR (qPCR), which has been extensively adopted as liquid biopsy technique in several cancer types, included CRC (see: 10.5301/ijbm.5000295, 10.3390/ijms19113356, 10.3390/cancers11101504). Please add a paragraph on qPCR, specifying pros and cons of this technique.

ddPCR: the adjective “inexpensive” is misleading, maybe you refer to the fact that is less expensive than NGS.

Lines 332-333 are unclear, please rephrase this period.

Lines 513-515 “In conclusion, in patients with stage III colon cancer, ctDNA may be used as a real-time marker of adjuvant therapy effectiveness. These observations open new opportunities for enriching recruitment of novel therapies in high-risk patients with detectable ctDNA”. I believe this phrase should be deleted from here and eventually moved to the end of the chapter. Moreover, not only novel therapies, but also new strategies adopting the already available drugs.

Line 569 please specify “colorectal” cancer recurrence

Lines 575-576 “However, serum CEA levels might only be elevated in 70-80% of patients” please add a reference here.

Line 631 please correct this typo: start and not star

Chapter 13: authors should cite also CIRCULATE-Japan trial 10.1111/cas.14926

Lines 680: not only more clinical trials are needed, but also data from ongoing ones

Round 2

Reviewer 2 Report

.